# The ellipse of insignificance, a refined fragility index for ascertaining robustness of results in dichotomous outcome trials

David Robert Grimes[1,2]*

[1]School of Physical Sciences, Dublin City University, Dublin, Ireland; [2]Discipline of Radiation Therapy, Trinity College Dublin, Dublin, Ireland

**Abstract** There is increasing awareness throughout biomedical science that many results do not withstand the trials of repeat investigation. The growing abundance of medical literature has only increased the urgent need for tools to gauge the robustness and trustworthiness of published science. Dichotomous outcome designs are vital in randomized clinical trials, cohort studies, and observational data for ascertaining differences between experimental and control arms. It has however been shown with tools like the fragility index (FI) that many ostensibly impactful results fail to materialize when even small numbers of patients or subjects in either the control or experimental arms are recoded from event to non-event. Critics of this metric counter that there is no objective means to determine a meaningful FI. As currently used, FI is not multidimensional and is computationally expensive. In this work, a conceptually similar geometrical approach is introduced, the ellipse of insignificance. This method yields precise deterministic values for the degree of manipulation or miscoding that can be tolerated simultaneously in both control and experimental arms, allowing for the derivation of objective measures of experimental robustness. More than this, the tool is intimately connected with sensitivity and specificity of the event/non-event tests, and is readily combined with knowledge of test parameters to reject unsound results. The method is outlined here, with illustrative clinical examples.

**\*For correspondence:**
davidrobert.grimes@dcu.ie

**Competing interest:** The author declares that no competing interests exist.

## Editor's evaluation

This valuable article describes a fragility index based on the geometry of chi-square tests. The result is linked to the concept of measurement error in outcomes, such that one can directly quantify how less-than-perfect sensitivity or specificity will call into question the statistical significance of a particular finding. The methodology rests upon solid mathematical exposition and several real-world examples of both interventional and observational studies. Noteworthy extensions for future considerations would be the application of this approach to censored outcomes.

## Introduction

Biomedical science is crucial for human well-being, but there is an increasing awareness that many published results are less robust than desirable (*Ioannidis, 2005*; *Loken and Gelman, 2017*; *Grimes et al., 2018*). In fields from psychology (*Krawczyk, 2015*) to cancer research (*Errington et al., 2021*), a substantial volume of research fails to replicate. There is an urgent need to address this, as spurious findings can not only obscure important research directions, but can even misinform potentially life-or-death decisions. While there are many reasons why published research might fail trustworthiness (including poorly conducted experiments, publish-or-perish pressure, and overt fraud in the form of data and image manipulation), inappropriate or misapplied statistical methods account for a large

**eLife digest** Science and medicine are vital to the well-being of humankind. Yet for all the incredible advances science has made, the unfortunate reality is that a worrying fraction of biological research is not reliable. Erroneous results might arise by chance or because of scientists' mistakes or ineptitude. Very occasionally, researchers may behave unethically and fabricate or inappropriately manipulate their data.

Inevitably, this can lead to untrustworthy research that misleads scientists and the public on questions integral to our health. Indeed, a recent study showed the results of several high-profile cancer papers could not be fully replicated. This problem is not unique to cancer, and studies on various other diseases have also not stood up to scrutiny from outside investigators. Finding ways to detect dubious results is therefore essential to protect the public's well-being and maintain public trust in science.

Here, Grimes demonstrates a new tool called the 'Ellipse of Insignificance' for measuring the reliability of dichotomous studies which are commonly used in many branches of biomedical sciences, including clinical trials. These studies typically compare two groups: one which was subjected to a specific treatment, and a control group which was not. Statistical methods are then applied to estimate how likely it is that differences in the number of observed events between the groups are real or due to chance.

The tool created by Grimes explores what would happen to seemingly strong results if some of the events in both the control and experimental arm of the study are recoded. It then assesses how much nudging is needed to change the statistical outcome of the experiment: the more interventions the result can withstand, the more robust the experiment. Grimes tested the tool and showed that a study suggesting a link between miscarriage and magnetic field exposure was likely unreliable because shifting the outcomes of less than two participants would change the result.

Scientists could use the Ellipse of Insignificance tool to quickly identify misleading published results or potential research fraud. Doing this could benefit researchers and protect the public from potential harm. It may also help preserve research integrity, increase transparency, and bolster public trust in science.

portion of misleading results. Even a properly performed statistical analysis may fail to adequately identify situations where data might lack robustness. p values are routinely misunderstood and misapplied, leading to confused research outputs (*Altman and Krzywinski, 2017*; *Colquhoun, 2014*; *Halsey et al., 2015*). Dichotomous outcome trials and studies are crucial in many avenues of biomedicine, from preclinical observational studies to randomized controlled trials. The essential principle is that they contrast experimental and control groups for some intervention, comparing the numbers positive for some specific endpoint in both arms. This is absolutely integral to modern medicine to ascertain significant differences, but some authors have voiced concern that seeming significant findings in these trials can often disappear with the recoding of even small numbers of patients from endpoint positive to negative in either arm. The fragility index (FI) is the measure of many subjects are required to change a trial outcome from statistical significance to not significant. It is calculated by recoding a patient or subject in the experimental group (or control group) from event to non-event, and employing Fisher's exact test until significance is lost. The number of patients requiring this recoding for this to occur is the FI. The concept of FI has existed in various forms since at least the work of *Feinstein, 1990*, and in general the higher the FI is, the more robust an experiment is deemed. Applications of FI have shown some concerning results; in a study of 399 randomized controlled trials (RCTs) in high-impact medical journals, *Walsh et al., 2014* found that median FI was 8 (range: 0–109), with 25% having FI $\leq$ 3. In 53% of these trials, numbers lost to follow-up exceeded FI. A meta-analysis of spinal surgery studies *Evaniew et al., 2015* found a median FI of 2, with 65% of trials having loss to follow-up greater than FI. A review of critical care trials (*Ridgeon et al., 2016*) and 2018 review of phase 3 cancer trials (*Del Paggio and Tannock, 2019a*) both found median FIs of 2, and a 2020 review of epilepsy research (*Das and Xaviar, 2020*) yielded a median FI of 1.5. A recent fragility analysis of COVID-19 trials found that had a median FI of only 4, despite the large numbers of patients involved (*Itaya et al., 2022*). This suggests that many results are not robust, and teeter on

the edge of statistical significance. While a very useful metric, FI has some substantial faults. There is considerable debate over whether is it appropriate for time-to-event cases (*Bomze and Meirson, 2019*; *Desnoyers et al., 2019*; *Machado et al., 2019*; *Del Paggio and Tannock, 2019b*). More directly, there is no simple FI cut-off metric that designates studies as either robust or fragile, though some authors suggest the fragility quotient (FQ) as an extension, the fraction of FI over sample size (*Tignanelli and Napolitano, 2019*). In addition, FI and FQ can also be computationally expensive to run, typically requiring multiple iterations of Fisher's exact test to converge. As Fisher's exact test relies on factorials, it is typically not suited to larger trials or studies. It is also implicitly considers only either control or experimental groups in isolation, even though it is possible that miscoding can occur in both cohorts. Nor does FI relate directly to test parameters between non-events and events, such as sensitivity or specificity. Many of these objections and counterpoints to them are discussed in recent work by *Baer et al., 2021a*. With FI and FQ becoming increasingly commonly reported in the literature, it is worthwhile to introduce a related, refined metric with new application. In this work, I introduce a geometric refinement of the concept underpinning FI which overcomes some difficulties associated with FI analysis, considering recoding in both control and experimental groups in tandem. This ellipse of insignificance (EOI) approach is exact and computationally inexpensive, yielding objective measures of experimental robustness. There are two major differences and situational advantages to such a formulation; firstly, it can handle huge data sets with ease and consider both control and experimental arms simultaneously, which traditional fragility analysis cannot. Previously, fragility has been typically considered in the case of relatively small numbers in RCTs, which as previous commentators have noted are often fragile by design. The method outlined here handles massive numbers with ease, rendering it suitable for analysis of observational trials, cohort studies, and general preclinical work, to detect dubious results and fraud. This sets it apart in both intention and application to existing measures, and makes it unique in this regard. Secondly, this methodology is not solely a new, robust FI; it also goes further by linking the concept of fragility to test sensitivity and specificity. This a priori allows an investigator to probe not only whether a result is arbitrarily fragile, but to truly probe whether consider certain results are even possible. This renders it less arbitrary than existent measures, as it ties directly statistically measurable quantities to stated results, and is sufficiently powerful to rule out suspect findings in many dichotomous trials and studies. It can accordingly be used to detect likely fraud or inappropriate manipulation of results if the statistical properties of the tests used are known. This is unfortunately highly relevant, as unsound or otherwise manipulated results have become an increasingly recognized problem in biomedical research, and means to detect them are vital. The EOI analysis outlined here for any $2 \times 2$ dichotomous outcome trial or study, with an experimental arm consisting of $a$ subjects with endpoint positive outcomes and $b$ without, and a control arm with $c$ subjects with endpoint positive versus $d$ without. The EOI analysis outlined in the methodology section allows rapid determination of the effects of recoding in all arms simultaneously, and ties this explicitly to test sensitivity and specificity, with illustrative examples of application demonstrated.

## Methods

The EOI approach is based upon the principles of a chi-squared analysis. Consider an experimental group containing $a$ participants with a given endpoint and $b$ participants without that endpoint. In the control group, there are $c$ participants with the given endpoint, and $d$ without. The total number of participants is given by $n = a + b + c + d$. For a 2 by 2 contingency table, the chi-squared statistic is given by

$$\chi_c^2 = \frac{n \left(ad - bc\right)^2}{(a + b)(c + d)(a + c)(b + d)}. \tag{1}$$

When this statistic is greater than a specified threshold, results are deemed significant and differences between the control and experimental groups considered indicative of real differences. The initial question this work concerns itself with is ascertaining how many patients or subjects would have to be recoded to transform an ostensibly significant result into one where the null hypothesis was not rejected. This recoding can be achieved two ways: by subtracting $x$ participants from $a$ (experimental

**Table 1.** Reported groups and related variables.

|  | **Endpoint positive** | **Endpoint negative** |
| --- | --- | --- |
| Experimental group | $a - x$ | $b + x$ |
| Control group | $c + y$ | $d - y$ |

group, endpoint positive) or by adding $y$ participants to $c$ (control group, endpoint positive). These configurations are given in *Table 1*.

Applying the same statistic outlined in *Equation 1*, with a threshold critical value for significance of $\nu_c$, the resulting identity is

$$\frac{n\Big((a-x)(d-y) - (b+x)(c+y)\Big)^2}{(a+b)(c+d)(a+c-x+y)(b+d+x-y)} - \nu_c = 0. \tag{2}$$

This form can be expanded, with the resultant equation being a conic section (*Grimes and Currell, 2018*) of the form $Ax^2 + Bxy + Cy^2 + Dx + Ey + F = 0$. This corresponds specifically to an inclined ellipse, with coefficients $A$–$F$ given by

$$A = (c+d)\big((c+d)n + (a+b)\nu_c\big) \tag{3}$$

$$B = 2(a+b)(c+d)(n - \nu_c) \tag{4}$$

$$C = (a+b)\big((a+b)n + (c+d)\nu_c\big) \tag{5}$$

$$D = (c+d)\big(2(bc - ad)n + (a+b)(b - a + d - c)\nu_c\big) \tag{6}$$

$$E = (a+b)\big(2(bc - ad)n + (c+d)(a - b + c - d)\nu_c\big) \tag{7}$$

$$F = (bc - ad)^2 n - (a+b)(a+c)(b+d)(c+d)\nu_c. \tag{8}$$

Any points on or in inside this EOI will fall below the threshold to reject the null hypothesis, and the ellipse is effectively the bound of all values of $x$ and $y$ sufficient to cause a loss of significance at a threshold critical value of $\nu_c$, calculated from the chi-squared distribution at a given level of significance with one degree of freedom.

## FECKUP point and vector

Finding the minimum distance from the origin to the EOI allows us to ascertain the minimal error which would render results insignificant. To find this, we take the implicit derivative of the distance vector from the origin to this unknown point, and the implicit derivative of the equation of the inclined ellipse whose coefficients are given in *Equations 3–8*. Setting $y'$ equal in both equations leads to the pair of simultaneous equations for the unknown point $(x_e, y_e)$ of

$$(2Ax_e + By_e + D)y_e - x_e(Bx_e + 2Cy_e + E) = 0 \tag{9}$$

$$Ax_e^2 + Bx_e y_e + Cy_e^2 + Dx_e + Ey_e + F = 0. \tag{10}$$

Solving this results in a quartic equation, resulting in four solutions, one pair of which will be the minimum distance point $(x_e, y_e)$. This can be readily checked, and the solution pair will correspond to the absolute minimum pair value to lose significance at a given threshold. This resultant point and vector denotes the Fewest Experimental/Control Knowingly Uncoded Participants (FECKUP), with length $f_{min}$. An illustration of this is shown in *Figure 1a*. Accordingly, the points $x_e$ and $y_e$ can be understood as the resolution of vector $f_{min}$ in the experimental and control directions, respectively. If both experimental and control participants can be miscoded, the theoretical minimum number that could be miscoded before a seemingly significant result dissipated, $d_{min}$, is the sum of the opposite and adjacent lengths of the right-angled triangle formed by hypotenuse $f_{min}$. As there are only integer numbers of participants, it thus follows that

$$d_{min} = \lfloor |x_e| + |y_e| \rfloor. \tag{11}$$

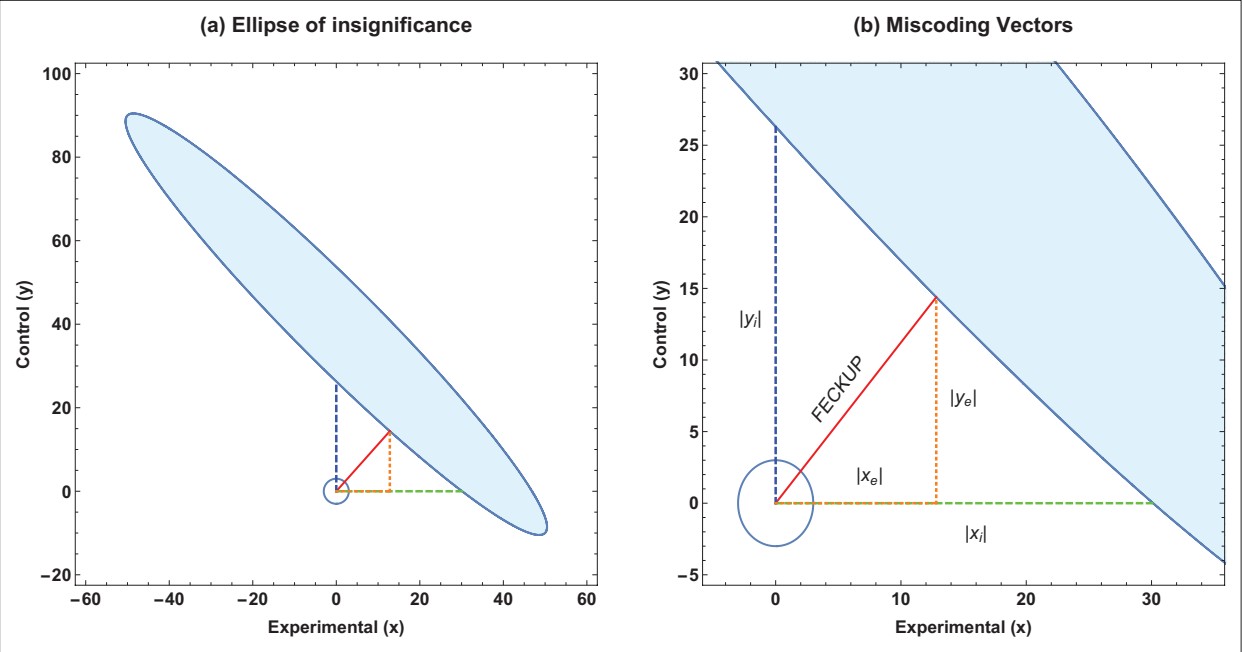

**Figure 1.** Ellipse of insignificance example. (**a**) An example ellipse of insignificance for the $a = 50$, $b = 50$, $c = 10$, $d = 90$ at a significance level of $\alpha = 0.05$. All points bounded by the ellipse depict $(x, y)$ combinations which would not lead to the null being rejected. (**b**) Relevant vectors for ascertaining misconding thresholds. In this example, the Fewest Experimental/Control Knowingly Uncoded Participants (FECKUP) point is $(x_e, y_e) = (12.8, 14.4)$, $f_{min} = 19.3$, and $(x_i, y_i) = (30.1, 26.3)$. See text for details.

If we instead only consider inaccuracies in the experimental group as possible, we may set $y = 0$ and $x = x_i$ for the equation of the ellipse, yielding the quadratic identity $Ax^2 + Dx + F = 0$, readily solvable to determine $x_i$. This is the point nearest the origin where the ellipse intercepts the $x$-axis. Conversely, we may consider a situation where only inaccuracies in the control group may exist. By similar reasoning, considering only inaccuracies in the control group yields a similar quadratic, $Cy^2 + Ey + F$ to yield $y_i$, the intercept of the ellipse with the $y$-axis. All these vectors are illustrated in *Figure 1b*, and are the maximum limits of miscoding theoretically possible before significance is lost.

## Metrics for fragility of results

To ascertain if a trial or study is robust against the miscoding of patients or subjects, we introduce metrics to quantify this. Considering only inaccuracies in the experimental group, we define the tolerance threshold for error in experimental group as the fraction of subjects that must be correctly allocated in the experimental group to maintain significance, given by

$$\epsilon_E = 1 - \frac{a + b - |x_i|}{a + b}. \tag{12}$$

This identity is intimately related to the existent FI, yielding the traditional FQ. For example, an experiment with $\epsilon_E = 0.1$ after EOI analysis would inform us that up to 10% of experimental participants could be miscoded before the result lost significance. By similar reasoning, the tolerance threshold for error allowable in the control group is then

$$\epsilon_C = 1 - \frac{c + d - |y_i|}{c + d}. \tag{13}$$

Finally, errors in both the coding of the experimental and control groups can be combined with FECKUP point knowledge. While $f_{min}$ gives a minimum vector distance to the ellipse, we instead take the length of the vector components to reflect to yield an absolute accuracy threshold of

$$\epsilon_A = 1 - \frac{n - |d_{min}|}{n}. \tag{14}$$

## Relating test sensitivity and specificity to miscoding thresholds

The identities derived thus far give a measure of the absolute accuracy required for confidence in the robustness of stated results. If details of the specific tests employed to determine endpoints in the experimental and control cohorts are known, then robustness can be directly related to the sensitivity and specificity of the tests employed. If the sensitivity ($s_{ne}$) and specificity ($s_{pe}$) of the test used to ascertain cases in the experimental group are known, then the observed number of cases with endpoint positive is related to the true number of endpoint positive cases, $a_o$, by $a = a_o s_{ne} + (a + b − a_o)(1 − s_{pe})$. It follows that the minimum miscoded cases in the experimental group are given by

$$x_m = \frac{b(1 − s_{pe}) − a(1 − s_{ne})}{s_{ne} + s_{pe} − 1}. \tag{15}$$

A similar relationship can be derived for the control groups, with sensitivity $s_{nc}$ and specificity $s_{pc}$, and the minimum miscoded cases in the control group are given by

$$y_m = \frac{c(1 − s_{nc}) − d(1 − s_{pc})}{s_{nc} + s_{pc} − 1}. \tag{16}$$

The values $(x_m, y_m)$ denote the minimum miscoding that exists in reported figures because of inherent test limitations, and it follows that if this pair value lies within the EOI, then any ostensible results of the study are not robust. The forms given in *Equations 15 and 16* are general forms. In many cases, when the same test is used in endpoint determination in the experimental and control groups, $s_{ne} = s_{nc}$ and $s_{pe} = s_{pc}$. However, there are instances when in observational and cohort trials in particular, accrued data will derive from different tests on various cohorts, an example of which will be introduced later in this work.

## Method inversion

It is important to note that the analysis presented here can be used not only to ascertain miscoding between endpoint positive and negative situations, but also can be inverted for situations where, for example, endpoint positive or negative might be known with high certainty but there are concerns over miscoding between control and experimental groups. In this case, simply reassigning endpoint

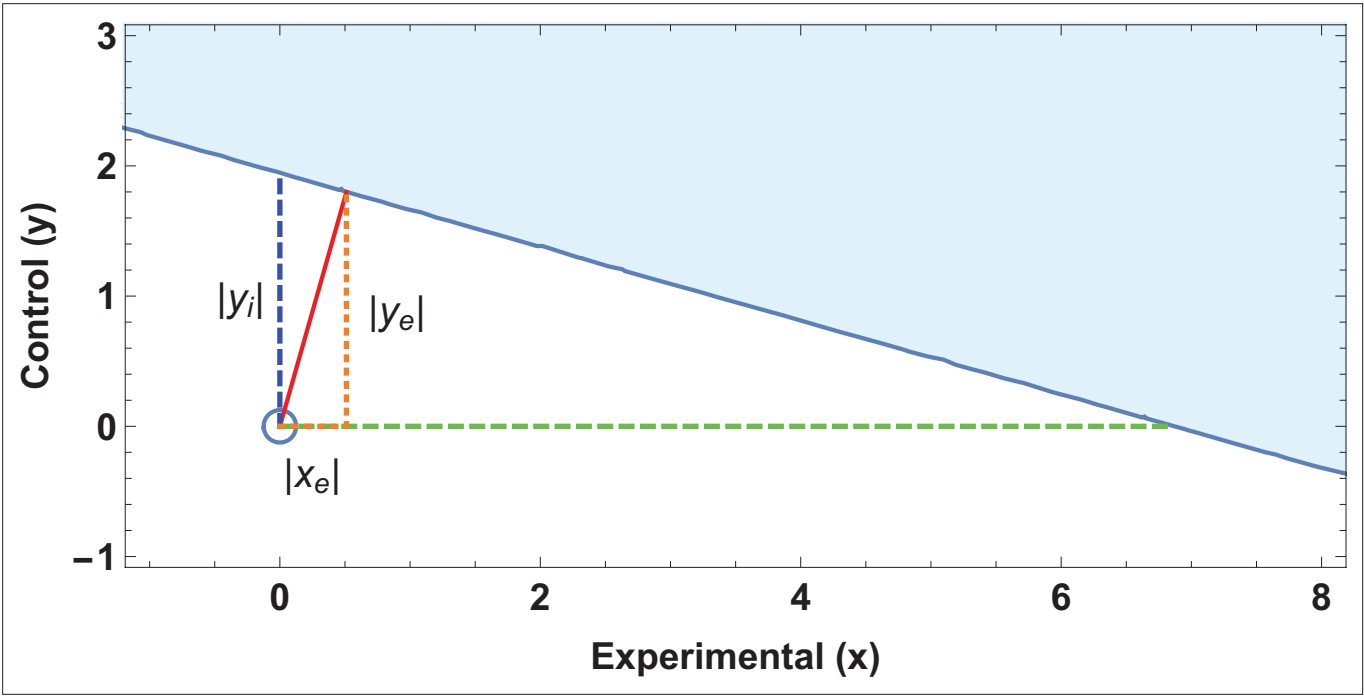

**Figure 2.** Application of ellipse of insignificance analysis to existent data. (**a**) Ellipses of insignificance analysis for a published study ($n$ = 913) for illustrative example 1 of published data. The shaded region denotes the ellipse of insignificance, the red line shows the Fewest Experimental/Control Knowingly Uncoded Participants (FECKUP) vector (the minimum vector from the origin to the ellipse).

**Table 2.** Ellipse of insignificance (EOI) derived metrics for published data.

| EOI statistic ($\alpha = 0.05$) | Derived value |
| --- | --- |
| Experimental group tolerance $x_i$ | 6.9 subjects |
| Control group tolerance $y_i$ | 1.9 subjects |
| FECKUP vector length | 1.9 subjects |
| Tolerance threshold for error (experimental group)$\epsilon_E$ | 0.99% |
| Tolerance threshold for error (control group)$\epsilon_C$ | 0.89% |
| Absolute tolerance threshold for error (all subjects)$\epsilon_A$ | 0.22% |

positive, experimental and control groups, respectively, as $(a, b)$ and endpoint negative experimental and control groups as $(c, d)$ allows straightforward application of EOI analysis as outlined.

## Polygon of insignificance

The EOI yields a continuously valued boundary. As only integer values are generally of concern, we can also define an irregular polygon of insignificance by considering the largest integer-valued polygon encompassing the EOI. Similarly, we can also take the floor values of $x_e, y_e, x_i$, and $y_i$ in such an approach. This is readily derived from EOI analysis, and code to produce such a shape is included in the supplementary material.

# Results
## Illustrative example 1 – EOI analysis of published data

A previously published study claimed higher rates of miscarriage in a cohort with high magnetic field exposure ($a = 164, b = 530$) versus a low exposure cohort ($c = 36, d = 183$), significant at $\alpha = 0.05$. An EOI analysis shows that a displacement of less than two subjects would be enough to undo this seeming significance as shown in *Figure 2*, and that the absolute tolerance threshold was only $\epsilon_A = 0.22\%$ as given in *Table 2*. This rendered the actual result highly fragile, given the demonstrable

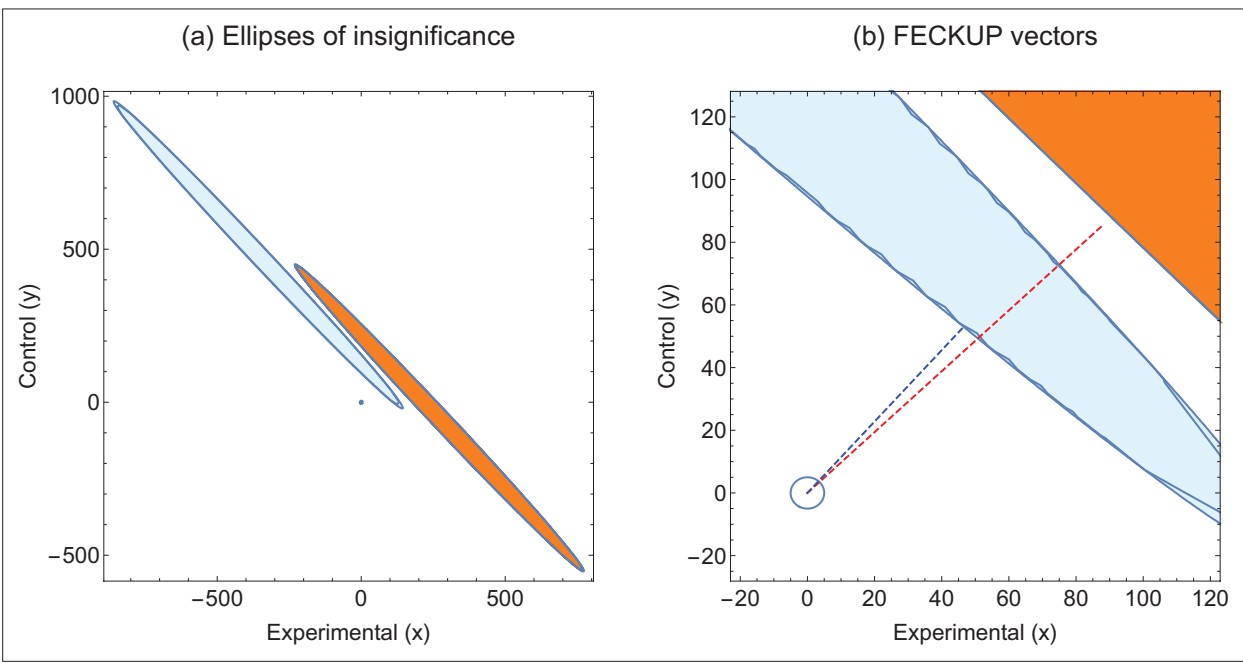

**Figure 3.** Illustrative example 2. (**a**) Ellipses of insignificance analysis for two studies with same $\chi^2$ statistic. (**b**) Fewest Experimental/Control Knowingly Uncoded Participants (FECKUP) vectors for both studies. Experiment 1 is given by orange ellipse and red dotted line, Experiment 2 by the blue ellipse and dotted line.

**Table 3.** Experimental metrics for similar test statistics.

| Significance level | Data | $\epsilon_E$ | $\epsilon_C$ | $\epsilon_A$ |
|---|---|---|---|---|
| $\alpha = 0.05$ | Experiment 1 | 17.7% | 18.2% | 8.9% |
| | Experiment 2 | 11% | 9.5% | 5.0% |
| $\alpha = 0.01$ | Experiment 1 | 16.3% | 17.0% | 8.3% |
| | Experiment 2 | 10.4% | 8.6% | 4.6% |
| $\alpha = 0.001$ | Experiment 1 | 14.8% | 15.5% | 7.5% |
| | Experiment 2 | 9.8% | 7.6% | 4.1% |
| $\alpha = 0.0001$ | Experiment 1 | 13.5% | 14.3% | 6.9% |
| | Experiment 2 | 9.2% | 6.8% | 3.8% |

fact that inspection of the supplied tables in the paper in question demonstrated that at least nine subjects had been miscoded in the initial analysis. These weaknesses, coupled with the lack of a plausible biophysical hypothesis and non-physical dose–response curve, suggests such findings were likely spurious (*Grimes and Heathers, 2021a*).

## Illustrative example 2 – EOI robustness analysis of similar results

Consider two hypothetical experiments that yield highly similar $\chi^2$ statistics. Experiment 1 has $(a_1, b_1, c_1, d_1) = (770, 230, 550, 450)$ and Experiment 2 gives $(a_2, b_2, c_2, d_2) = (144, 856, 20, 980)$, both of which correspond to $\chi^2 \approx 100$, and p values <0.00001. We can employ EOI analysis to ascertain how robust these seemingly strong respective results are for different values of $\alpha$. The EOI analysis and FECKUP vectors are illustrated in *Figure 3* for $\alpha = 0.05$, and relevant statistics for various values of $\alpha$ are given in *Table 3*. It can be seen from this that despite the similar test statistics, Experiment 1 is consistently more robust, and would require the miscoding of at least 178 participants (8.9% of the entire sample) to lose significance, relative to 99 ($\approx$ 5% of the entire sample) in Experiment 2 at $\alpha = 0.05$, a trend that continues even with lower values of $\alpha$.

## Illustrative example 3 – sensitivity and specificity in cancer screening statistics

Consider an application of EOI analysis where sensitivity and specificity of different tests are being implicitly compared. Screening results derived from two hypothetical cities are listed in *Table 4*. City A uses standard Liquid-based cytology (LBC) analysis whereas City B's programme uses a HPV(human papillomavirus)reflex scheme, where subjects are first tested for high-risk HPV. With $p < 0.00001$, it would seem highly significant that these two cities have markedly different rates of CIN2+. The EOI analysis reveals that FECKUP vector details, as shown in *Figure 4*. Yet as the sensitivity and specificity of the respective tests are known (LBC: $s_n = 0.75$, $s_p = 0.90$, HPV-reflex: $s_n \approx 0.68$, $s_p \approx 0.99$) application of *Equation 14* yields $x_m \approx 93$. This exceeds $x_i$ and lies within the EOI, meaning we can immediately discount the ostensibly highly significant result despite its seeming strength. Further application of EOI analysis informed by sensitivity and specificity allows us to ascertain that the two cities actually have the same prevalence of CIN2, at 20 cases per 1000, a real problem encountered when comparing national screening programmes (*Grimes et al., 2021c*).

**Table 4.** Results of different analysis.

| | CIN2 + positive | No CIN2 + detected | Methodology |
|---|---|---|---|
| City A (measured) | 113 | 887 | LBC only |
| City B (measured) | 24 | 976 | HPV screening/LBC reflex |
| True values (both cities) | 20 | 980 | N/A |

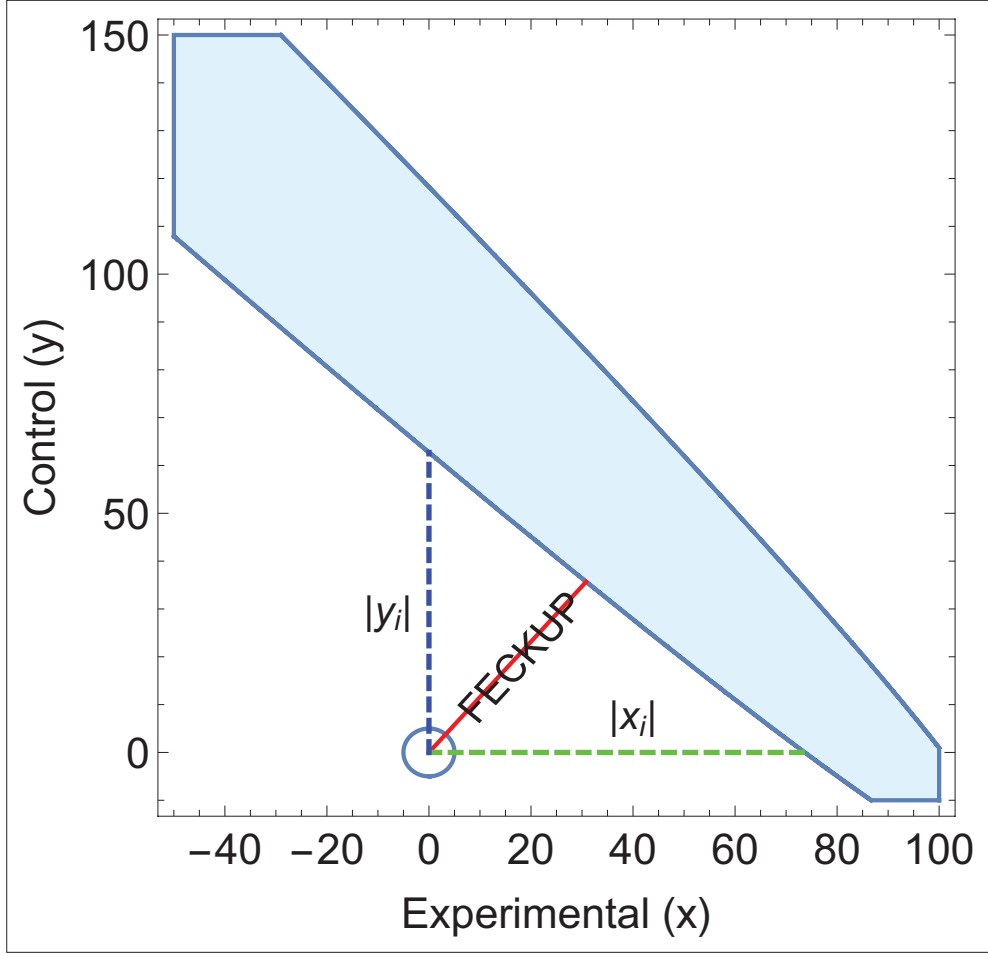

**Figure 4.** Illustrative example 3. An ellipse of insignificance (EOI) analysis on the data supplied in the City A/City B screening comparison yields a Fewest Experimental/Control Knowingly Uncoded Participants (FECKUP) vector (in red) of 46.2 subjects, corresponding to a minimum tolerance of 66.5 total subjects after resolving the vector. As $x_i = 73.7$ (shown in green) with $y_i = 62.7$ (shown in blue), but as the sensitivity and specificity of the tests used in City A are known, it can be shown that $x_m \approx 93$, exceeding the limits of $x_i$, placing the point within the ellipse and rendering any seeming significance void. Note that only a part of the EOI (denoted by the blue solid shape) is shown for clarity.

## Discussion

The analysis presented here is a deterministic way to ascertain the fragility of a given dichotomous outcome study by considering experimental and control groups in concert. This method is geometrical in origin and computationally inexpensive. It also explicitly can relate outcome fragility to the sensitivity and specificity of tests employed when known, aiding clinicians and meta-researchers in interpreting the trustworthiness of a given study. Sample OCTAVE and MATLAB code and stand-alone *Windows* applications are provided to run the analysis outlined in this work, available in the electronic supplementary material. There are a number of limitations of this work that should be explicitly discussed, and caveats to be elucidated. The EOI analysis handles potential miscoding, but cannot be used to infer anything about patients or subjects lost to follow-up. This is a weakness of all FI/FQ methods, as it is not a priori knowable from reported data alone why patients dropped out, or why they might have atrophied from particular subgroups. Redaction bias (***Grimes and Heathers, 2021b***) can occur if subjects leave a particular subset at an elevated rate, and while beyond the scope of this work, it is important to realize that explicit connections between EOI/FI/FQ analysis and numbers lost to follow-up cannot be directly made. The method outlined is deterministic and rapid, but only currently applicable to dichotomous outcome trials and studies, and should be applied very cautiously to time-to-event data, where it may not be suitable. FI itself is also typically calculated using Fisher's

exact test, which well approximates a chi-squared test. However, for small trials, the p value derived from Fisher's exact test can be discrepant from chi-squared result. When Fisher's exact test produces a non-significant p value without any recoding, an FI of 0 results, suggesting a distinct lack of robustness of the underlying data. As EOI analysis is built upon chi-squared statistics, it is possible in edge cases of small numbers to have discordant results between EOI and Fisher's exact test also. The chief advantage of the method outlined here, however, is that it handles extremely large data sets with ease. In large data sets, Fischer's exact test breaks down due to its dependence on factorials, and a chi-squared approximation is more appropriate. This is fitting, given EOI is built upon the chi-square distribution. But the important caveat is that for rare events in small trials, an FI approach built upon Fisher's exact test may be more appropriate (*Baer et al., 2021a*). The usage of FI/FQ itself remains contested in the literature, and one frequent objection is that the mere existence of a small FI might be an artefact of trial design (*Walter et al., 2020*). With clinical RCTs in particular, experimenters often design trials to minimize exposure of patients or subjects to as of yet unknown harms, while seeking to ensure enough of them participate so that clinically relevant causal effects can be reliably detected. From this vantage point, RCTs might be fragile 'by design'. This view is countered by other authors *Baer et al., 2021a* who argue that there are no evidence p value distributions tend to cluster around the significance threshold after a sample size calculation, and that the FI in well-designed studies is not always low (*Baer et al., 2021b*). This work does not comment on the absolute applicability of the FI, but offers new metrics for quantification of results in context. More importantly, EOI analysis has definite application for dichotomous outcome results not derived just from fragile-by-design RCTs, but from ecological studies, cohort trials, and preclinical work which should in principle be far more resilient to investigation than RCTs. There is a less edifying but important reason why EOI analysis might be conducted – the detection of questionable research practices and fraud. While most scientists and clinicians operate ethically, poor conduct and inappropriate statistical manipulation can and do occur. By some estimates, up to three quarters of all biomedical science are affected by poor practice (*Fanelli, 2009*), casting doubt on results to the detriment of science and the public, often a consequence of publish-or-perish pressure (*Grimes et al., 2018*). During the COVID-19 pandemic, a number of dubious high-profile results have come to light, particularly on drugs like Ivermectin (*Hill et al., 2022*; *Besançon et al., 2022*). EOI analysis has a potential role in detecting manipulations that nudge results towards significance, and identifying inconsistencies in data. EOI analysis is perhaps ideal for this purpose, as it explicitly relates known test sensitivity and specificity to projected error tolerance, allowing detection of suspect results in even large data sets, as illustrated by the real examples in this work. Despite its caveats on usage, the FI has seen growing application in analysis of trial outcomes, and the EOI system presented here should allow this to be applied more thoroughly in a multidimensional way. Regardless of whether appropriate research practice has been observed or not, it is important to be able to estimate the soundness of results in biomedical science, to ascertain what level of confidence once can ascribe to them. This need has seen the recent resurgence of FI analysis, and the EOI analysis presented here can help undercover questionable results and experimental inconsistencies, with wide potential application in meta-research and reproducible research.

## Acknowledgements

DRG thanks the Wellcome trust for their support, Dr Darren Dahly for insight into aspects of trial design, Dr Nick Brown for his proofing, and Dr James Heathers for encouragement, discussions, and creative profanity. He would also like to thank Dr Ben Baer, Dr Martin T Wells, and Faheem Gilani for their helpful comments on this manuscript. He could also like to extend this thanks to the reviewers for their diligence and advice.

## Additional information

### Funding

| Funder | Grant reference number | Author |
| --- | --- | --- |
| Wellcome Trust | 214461/A/18/Z | David Robert Grimes |

| Funder | Grant reference number | Author |
|---|---|---|

The funders had no role in study design, data collection, and interpretation, or the decision to submit the work for publication. For the purpose of Open Access, the authors have applied a CC BY public copyright license to any Author Accepted Manuscript version arising from this submission.

## Author contributions
David Robert Grimes, Conceptualization, Software, Formal analysis, Funding acquisition, Validation, Investigation, Visualization, Methodology, Writing - original draft, Project administration

## Author ORCIDs
David Robert Grimes http://orcid.org/0000-0003-3140-3278

## Decision letter and Author response
Decision letter https://doi.org/10.7554/eLife.79573.sa1
Author response https://doi.org/10.7554/eLife.79573.sa2

# Additional files

## Supplementary files
• MDAR checklist

• Source code 1. Sample MATLAB/OCTAVE code for rapid implementation of EOI analysis method outlined.

## Data availability
The paper is a modelling study and methodology and contains no data, and code provided in the supplementary material allows reproduction of all methods.

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
