## [Editor Report]

This valuable article describes a fragility index based on the geometry of chi-square tests. The result is linked to the concept of measurement error in outcomes, such that one can directly quantify how less-than-perfect sensitivity or specificity will call into question the statistical significance of a particular finding. The methodology rests upon solid mathematical exposition and several real-world examples of both interventional and observational studies. Noteworthy extensions for future considerations would be the application of this approach to censored outcomes.

---

## [Decision Letter]

**Decision letter after peer review:**

[Editors’ note: the authors submitted for reconsideration following the decision after peer review. What follows is the decision letter after the first round of review.]

Thank you for submitting the paper "The Ellipse of Insignificance: a refined fragility index for ascertaining robustness of results in dichotomous outcome trials" for consideration by *eLife*. My apologies for the delay in getting these reviews returned to you. Your article has been reviewed by 2 peer reviewers, including Philip Boonstra as the Reviewing Editor and Reviewer #1, and the evaluation has been overseen by a Senior Editor. The following individual involved in review of your submission has agreed to reveal their identity: Fei Jiang (Reviewer #2).

Comments to the Authors:

We are sorry to say that, after consultation with the reviewers, we have decided that this work will not be considered further for publication by *eLife*.

Specifically, as the reviewers note, the proposed approach does not adequately address the limitations of the existing family of fragility indices and therefore seems to suffer from the same limitations as the methods it is intended to improve upon, and the improvement over the fragility index is neither characterized or quantified.

*Reviewer #1 (Recommendations for the authors):*

This article extends the concept of the fragility index for clinical trials using geometric arguments. The proposed approach is a two-step calculation. First, the two-dimensional ellipsis that contains the insignificance region is calculated, where each dimension represents one of the two arms in the trial. Then, the shortest vector between the trial's actual result and this ellipsis is identified. Shorter-length vectors point to greater fragility in the findings.

The idea of the fragility index is to measure how many subjects' outcomes would need to be changed in order to change the qualitative conclusion of a trial. The author raises several challenges to the fragility index: a lack of feasibility for time-to-event outcomes, a lack of clear distinction between 'robust' versus 'fragile', the need to calculate Fisher's exact test multiple times, and, in some cases, an inability to deal with fragility in both treatment arms at the same time.

Unfortunately, the proposed idea, although mathematically very intriguing, does not ultimately address these stated deficiencies, which limits its utility. Specifically, there is no solution proposed for the issue of time-to-event outcomes; there is no resolution to the issue of distinguishing between robust and fragile; and the perceived computational burden of calculating multiple Fisher's exact tests is actually not particularly great given the computational power of today's personal computers. Thus, the primary contribution of this article – from this reviewer's perspective – is with regard to the ability to generalize the fragility index to considering changes in both arms simultaneously.

Separately – I wonder if it would be more appropriate to define an irregular polygon of insignificance, defined as the largest polygon that is encompassed by the EOI and is comprised of connected segments of integer valued (x,y) coordinates. My thinking here is that since it is impossible to have continuously valued counts of responses, one should only consider integer-valued (x,y) coordinates as possibilities.

It may also be worthwhile to compare and contrast the difference in assumptions and approximations between Fisher's exact test and a chi-squared analysis.

1. The chosen acronym (FECKUP) is very similar-sounding and similar-looking to an English-language vulgarity. I wonder if the author might consider a different choice of acronym for their index.

2. The definitions of Qe, Qc, and Qa in Inequalities (11), (12), and (13), respectively, are somewhat unclear. Since these are inequalities, the mathematical implication is that Qe, Qc, and Qa can be any values less than their upper bounds, as in rows 4 and 5 of Table, whereas they are presented as single values in Table 3. I think the intent is to define Qe, Qc, and Qa as being equal to these upper bounds and to then claim that these are 'best case' scenarios for these different error types. Can the author please clarify/change how precisely to interpret Qe, Qc, and Qa and potentially consider changing these expressions in the manuscript?

*Reviewer #2 (Recommendations for the authors):*

This paper provides a potentially useful tool for conducting reproducible research with binary endpoints. The work provides steps to construct evaluation criteria of how sensitive a scientific conclusion regarding the changes of the observations. Overall, the study problem is important, but the proposed methods are not fully evaluated against the existing approach to justify the superiority of the method.

Strengths:

The derivation of the measurement is comprehensive.

Two case studies are interesting

Weaknesses:

No comparison with the existing method.

The author claim that the method is computationally inexpensive, but did not report the computational time for the proposed method and the existing method.

It is not clear what the improvement the proposed method made upon the existing method.

Impact:

The new tool can be useful for conducting reproducible research if the methods have been validated more rigorously.

Suggestions:

1, The paper will benefit from adding some simulation studies to justify the superiority of the method.

2, The author should be more precise about their terminologies. For example, what does "minimal error", "maximal proportion error" refer to? Also (11), (12), and (13) only provide upper bound of Qe, Qc, Qa, not their exact values. It is not clear why Qa reported in Table 2 has an exact solution. Please clarify.

3, The authors must compare with existing methods in real data examples.

[Editors’ note: further revisions were suggested prior to acceptance, as described below.]

Thank you for resubmitting your work entitled "The Ellipse of Insignificance, a refined fragility index for ascertaining robustness of results in dichotomous outcome trials" for further consideration by *eLife*. Your revised article has been evaluated by Mone Zaidi (Senior Editor) and a Reviewing Editor. Thanks also for your patience in getting this review back to you.

The manuscript has been improved but there are some remaining issues that need to be addressed, as outlined below.

*Reviewer #1 (Recommendations for the authors):*

Thanks to the author for their revised submission. Now that I have a better understanding of the approach, I do unfortunately have some additional questions, not all of which were raised in my initial review.

1. I am confused by the difference in definition between x_i vs x_m and y_i vs y_m. The legend in Figure 2 states that the green line depicts x_i, whereas the caption in Figure 2 caption states that the green line depicts x_m. Similarly, Table 1 also implies that x_m is the green line by virtue of stating that it is equal to 6.9. In fact, x_i is not explicitly defined in the manuscript as far as I can tell, but based upon the paragraph before equation (11), I believe it is interpreted as 'the number of subjects in the experimental group with a recorded positive endpoint who would need to be reclassified to negative in order for the study to lose statistical significance at the given significance threshold, holding fixed all measurements in the control group'. And y_i would be defined analogously. I think these are more or less the classic FI metrics. And therefore, I believe that the green line in Figure 2 is x_i, not x_m. In contrast, x_m and y_m must take into account some external knowledge/assumptions about the sensitivity and specificity of the measurement process; these values cannot be learned from the data itself. The sentence just before equations (14) and (15) states that x_m and y_m are the 'minimum miscoded cases', but I believe they are more appropriately defined as an 'anticipated number of miscoded cases given presumed values of specificity and sensitivity' (in particular, I would argue that they should be written as functions of se and sp: xm(se, sp)). If my understanding about this is all correct, I do not believe these definitions are adequately communicated in the manuscript and sometimes contradictorily used (e.g. in the caption of Figure 2). I would suggest, for example, that in the illustrative example on the bottom of page 4, it is clearly stated what x_i and y_i are, what the presumed values of se and sp are, and therefore what the presumed values of x_m and y_m are (as a related aside: how is the reader supposed to know that it is demonstrable fact that at least 9 patients had been miscoded? Is this discussed in the reference?).

2. Thanks to the author for clarifying the definitions of Qe, Qc, and Qa (now defined as epsilon_E, etc). I especially appreciate the added sentence between equations (11) and (12) that interpret it for a non-statistical audience. However, I see that the Methods section has been moved after the Results and Discussion section. I am not strictly opposed to this decision; however, the result is that readers who read through the manuscript as it is written will see these novel technical terms used prior to their definitions. I suggest that the author either return to the ordering of sections that is more traditionally used for statistical articles and which was used in the original submission (Introduction, Methods, Results, Discussion) or add references to defining equations for any novel technical terms used at their first use and a simple, non-statistical interpretation.

3. If the definitions /distinctions between x_i and x_m can be cleared up as per my first comment above, are the epsilons really necessary at all? Put differently, can the author please clarify why these statistics offer distinct information? Epsilon_A would seem to be just a rescaling of x_i by the sample size.

4. Figure 4: I suggest to add a parenthetical that the FECKUP vector is in red. A legend such as what is used in Figure 2 would be helpful.

5. Also, in the spirit of creating Figures that 'stand alone', can the caption of Figure 4 be modified to make clear which illustrative example it refers to? The only evidence that Figure 4 refers to the hypothetical liquid biopsy data is the use of the term 'city A' in the second sentence of the caption.

6. Figure 2 is not referenced anywhere in the text of the manuscript (as far as I can tell).

7. Can the author please add some sort of enumerative label or title to each of the illustrative examples and ensure that tables and figures make clear in their captions which illustrative example they refer to?

8. The third paragraph of the discussion says the method is 'only currently applicable to dichotomous outcome trials' but, assuming that the word 'trial' refers to an active intervention, I believe the author intends that this approach is more widely applicable to any study (that is, both interventional and observational) with a non-censored dichotomous endpoint.

9. I am still unclear on the need for the inequality in (13): does this inequality give a definition of epsilon_a, i.e. is epsilon_a really the entire set of numbers less than the RHS of this inequality? If so, why is epsilon_a only ever given as a single number and not an interval?

10. Figure 4: presuming this Figure refers to the cancer screening example, the units of analysis here are more appropriately referred to as 'subjects' not 'patients'. This comment applies more broadly, i.e. the second paragraph of the discussion.

11. The statement immediately following (15): "If xm >= xi or ym >= yi or both conditions are met…". Should there be absolute values around these expressions? For example, if xi < 0 and xm=0, then the sentence as currently written suggests that the results are not robust, which seems misleading. I believe the intended meaning here is "If |xm| >= |xi| or |ym| >= |yi| or both conditions are met…".

---

## [Author Response]

[Editors’ note: The authors appealed the original decision. What follows is the authors’ response to the first round of review.]

Reviewer #1 (Recommendations for the authors):This article extends the concept of the fragility index for clinical trials using geometric arguments. The proposed approach is a two-step calculation. First, the two-dimensional ellipsis that contains the insignificance region is calculated, where each dimension represents one of the two arms in the trial. Then, the shortest vector between the trial's actual result and this ellipsis is identified. Shorter-length vectors point to greater fragility in the findings.The idea of the fragility index is to measure how many subjects' outcomes would need to be changed in order to change the qualitative conclusion of a trial. The author raises several challenges to the fragility index: a lack of feasibility for time-to-event outcomes, a lack of clear distinction between 'robust' versus 'fragile', the need to calculate Fisher's exact test multiple times, and, in some cases, an inability to deal with fragility in both treatment arms at the same time.1. Unfortunately, the proposed idea, although mathematically very intriguing, does not ultimately address these stated deficiencies, which limits its utility. Specifically, there is no solution proposed for the issue of time-to-event outcomes; there is no resolution to the issue of distinguishing between robust and fragile; and the perceived computational burden of calculating multiple Fisher's exact tests is actually not particularly great given the computational power of today's personal computers. Thus, the primary contribution of this article – from this reviewer's perspective – is with regard to the ability to generalize the fragility index to considering changes in both arms simultaneously.

This is a very helpful comment and raises important aspects I’ll address here. Firstly, part of the problem is that I did not clarify the true motivation for this work: to not only consider relatively small RCTs but to create a robust framework for observational, longitudinal, cohort, and preclinical trials with dichotomous outcomes, tied directly to the properties of the statistical tests used to create that dictomization in the first instance. While this method doesn’t resolve time-to-event problem, it is capable of handling huge trials where Fischer’s exact test would struggle or fail due to its inherent reliance on factorials. This makes it rugged and capable of handling much more than RCTs. And more than this, because of the equations 14 and 15, we can in many cases directly contrast the explicit fragility with the theoretical thrust limits given by the tests involved, which will allow us to rule out many stated results without having to arbitrarily set a threshold, as discussed above. This makes it highly suitable for automated meta-analysis, and detection of fraud.

I have accordingly re-written the introduction and discussion text to reflect this motivation better, with the modified passages now reading:

“In this work, I introduce a geometric refinement of the concept underpinning FI which overcomes some difficulties associated with FI analysis, considering recoding in both control and experimental groups in tandem. This ellipse of insignificance (EOI) approach is exact and computationally inexpensive, yielding objective measures of experimental robustness. There are two major differences and situational advantages to such a formulation; firstly, it can handle huge data sets with ease and consider both control and experimental arms simultaneously, which traditional fragility analysis cannot. Previously, fragility has been typically considered in the case of relatively small numbers in Randomized controlled trials, which as previous commentators have noted are often fragile by design. The method outlined here handles massive numbers with ease, rendering it suitable for analysis of observational trials, cohort studies, and general preclinical work, to detect dubious results and fraud. This sets it apart in both intention and application to existing measures, and makes it unique in this regard.

Secondly, this methodology is not solely a new, robust fragility index; it also goes further by linking the concept of fragility to test sensitivity and specificity. This {a priori} allows an investigator to probe not only whether a result is arbitrarily fragile, but to truly probe whether consider certain results are even possible. This renders it less arbitrary than existent measures, as it ties directly statistically measurable quantities to stated results, and is sufficiently powerful to rule out suspect findings in many dichotomous trials. It can accordingly be used to detect likely fraud or inappropriate manipulation of results if the statistical properties of the tests used are known. This is unfortunately highly relevant, as unsound or otherwise manipulated results have become an increasingly recognised problem in biomedical research, and means to detect them are vital.”

2. Separately – I wonder if it would be more appropriate to define an irregular polygon of insignificance, defined as the largest polygon that is encompassed by the EOI and is comprised of connected segments of integer valued (x,y) coordinates. My thinking here is that since it is impossible to have continuously valued counts of responses, one should only consider integer-valued (x,y) coordinates as possibilities.

This is an excellent suggestion and echoes one made to me by readers of the preprint. I have already created a method for precisely doing this which compliments the existing ellipse method, and this is now incorporated into the manuscript and the sample code. The results are broadly similar to the EOI method, but it is a useful addition to consider. The new text reads:

“The ellipse of insignificance yields a continuously valued boundary. As only integer values are generally of concern, we can also define an irregular polygon of insignificance (POI) by considering the largest integer-valued polygon encompassing the EOI. Similarly, we can also take the floor values of xe, ye ,xi, and yi in such an approach. This is readily derived from EOI analysis, and code to produce such a shape is included in the supplementary material.”

3. It may also be worthwhile to compare and contrast the difference in assumptions and approximations between Fisher's exact test and a chi-squared analysis.

This is an excellent idea too to elucidate differences. As I have clarified now in reply point 1 about the application being different, it’s worth stressing the differences again for comparison. In the discussion, I’ve added the text:

“As EOI analysis is built upon chi squared statistics, it is possible in edge cases of small numbers to have discordant results between EOI and Fisher's exact test also. The chief advantage of the method outlined here, however, is that it handles extremely large data sets with ease. In large data sets, Fischer's exact test breaks down due to its dependence on factorials, and a chi-squared approximation is more appropriate. This is fitting, given EOI is built upon the chi-square distribution.

But the important caveat is that for rare events in small trials, a Fragility index approach built upon Fisher's exact test may be more appropriate.”

4. The chosen acronym (FECKUP) is very similar-sounding and similar-looking to an English-language vulgarity. I wonder if the author might consider a different choice of acronym for their index.

This is true and partly tongue-in-cheek; in Hiberno and British English ‘feck’ is a sanitised and more gentle version of the more common vulgarity, and I chose it here in part because the natural acronym that arose alluded to the idea of messing something up without the aggressive qualities of the more Germanic term. I am completely happy to change this if the reviewer and editors wish, and am happy to act on their advice and change or leave it as requested.

5. The definitions of Qe, Qc, and Qa in Inequalities (11), (12), and (13), respectively, are somewhat unclear. Since these are inequalities, the mathematical implication is that Qe, Qc, and Qa can be any values less than their upper bounds, as in rows 4 and 5 of Table, whereas they are presented as single values in Table 3. I think the intent is to define Qe, Qc, and Qa as being equal to these upper bounds and to then claim that these are 'best case' scenarios for these different error types. Can the author please clarify/change how precisely to interpret Qe, Qc, and Qa and potentially consider changing these expressions in the manuscript?

I think this is a very fair criticism that has been echoed by preprint comments, and it is apparent that my initial formulation lacked clarity. I have totally rewritten this section (and all related figures) and also altered the metrics slight to be more intuitive, and instead defined the quantities as tolerances. For example, I know define *ε*_E_ as the tolerance threshold for error in the experimental group; for example, if *ε*_E_ = 0.1, that is the same as observing that 10% of the control group could be recoded from event to non-event (or vice versa) before seeming significant was lost. This tolerance is more intuitive, and I’ve changed the metric throughout: see ‘Metrics for fragility of results’:

Reviewer #2 (Recommendations for the authors):1. This paper provides a potentially useful tool for conducting reproducible research with binary endpoints. The work provides steps to construct evaluation criteria of how sensitive a scientific conclusion regarding the changes of the observations. Overall, the study problem is important, but the proposed methods are not fully evaluated against the existing approach to justify the superiority of the method.

The major strength of this method lies in ruggedness of the method to huge trials (not just RCTs) and its ability to relate results to the sensitivity / specificity of the actual tests used to dictomize the data. It is accordingly useful for fraud detection, and for identifying weak results without the same arbitrariness of other metrics in many cases. Please see replies 1 and 3 to reviewer 1.

Suggestions:1, The paper will benefit from adding some simulation studies to justify the superiority of the method.2, The author should be more precise about their terminologies. For example, what does "minimal error", "maximal proportion error" refer to? Also (11), (12), and (13) only provide upper bound of Qe, Qc, Qa, not their exact values. It is not clear why Qa reported in Table 2 has an exact solution. Please clarify.

This is absolutely true, and resonates with comments by reviewer 1 point 5. I have made these much more solid and intuitive in the current iteration, and please see reply to reviewer 1 point 5 for specific details.

[Editors’ note: what follows is the authors’ response to the second round of review.]

The manuscript has been improved but there are some remaining issues that need to be addressed, as outlined below.Reviewer #1 (Recommendations for the authors):Thanks to the author for their revised submission. Now that I have a better understanding of the approach, I do unfortunately have some additional questions, not all of which were raised in my initial review.1. I am confused by the difference in definition between x_i vs x_m and y_i vs y_m. The legend in Figure 2 states that the green line depicts x_i, whereas the caption in Figure 2 caption states that the green line depicts x_m. Similarly, Table 1 also implies that x_m is the green line by virtue of stating that it is equal to 6.9. In fact, x_i is not explicitly defined in the manuscript as far as I can tell, but based upon the paragraph before equation (11), I believe it is interpreted as 'the number of subjects in the experimental group with a recorded positive endpoint who would need to be reclassified to negative in order for the study to lose statistical significance at the given significance threshold, holding fixed all measurements in the control group'. And y_i would be defined analogously. I think these are more or less the classic FI metrics. And therefore, I believe that the green line in Figure 2 is x_i, not x_m. In contrast, x_m and y_m must take into account some external knowledge/assumptions about the sensitivity and specificity of the measurement process; these values cannot be learned from the data itself. The sentence just before equations (14) and (15) states that x_m and y_m are the 'minimum miscoded cases', but I believe they are more appropriately defined as an 'anticipated number of miscoded cases given presumed values of specificity and sensitivity' (in particular, I would argue that they should be written as functions of se and sp: xm(se, sp)). If my understanding about this is all correct, I do not believe these definitions are adequately communicated in the manuscript and sometimes contradictorily used (e.g. in the caption of Figure 2). I would suggest, for example, that in the illustrative example on the bottom of page 4, it is clearly stated what x_i and y_i are, what the presumed values of se and sp are, and therefore what the presumed values of x_m and y_m are (as a related aside: how is the reader supposed to know that it is demonstrable fact that at least 9 patients had been miscoded? Is this discussed in the reference?).

I thank the reviewer for raising this, as it lays bare how clumsy I had been with my terminology, partially a consequence of multiple iterations getting confused. I have amended the text and images very substantially in this revision. Briefly, if we consider the ellipse of insignificance, the FECKUP vector is the closest line from the origin to this ellipse. The resolution of this vector gives us (x_m_, y_m_), the absolute minimum number of experimental and control subjects respectively would have to be miscoded for significance to vanish. The value (x_i_) arises from a hypothetical situation where only experimental subjects can vary (we have perfect accuracy in control subjects), which corresponds to the ellipse intersecting the x-axis. Conversely, we have (y_i_), the hypothetical situation where the experimental arm is perfectly accurate, but the control arm can vary. This is where the ellipse intersects the y-axis. In the new 2-part figure 1, I have now hopefully made this much clearer. These points allow us to state the maximum tolerance for miscoding we can even hypothetically have, bounded between the two extremes. I hope that the new figure 1(b) remedies this somewhat.

The situation for (x_m_,y_m_) was also poorly elucidated by me originally – my apologies, I was very unclear in my phrasing. Put simply, if a test for sorting between events and non-events is not perfect but has some known sensitivity and specificity, then it follows that the reported numbers will be affected by this. The identities established in this work allow us to work backwards, and explicitly determine how many cases must be miscoded for a known test sensitivity / specificity. We can then derive this pair from the equations outlined, and check whether (x_m_, y_m_) lies within the ellipse of insignificance. If it does, as in the cancer screening example, we can confidently state that any seeming significance is entirely illusory. I was clumsy in this phrasing and contradictory, and have now rewritten these sections for clarity. It was confusing to relate these to (x_i_) because, more intuitively, all we need to do so show a result is not robust is show (x_m_,y_m_) lies within the ellipse of insignificance. I have rewritten the section now to account for this, explicitly stating

“The values for (x_m_, y_m_) denote the minimum miscoding that exists in reported figures because of inherent test limitations, and it follows that if this pair-value lies within the ellipse of insignificance, then any ostensible results of the study are not robust.”

Hopefully this clarifies somewhat – in relation to the figures, the reviewer is absolutely correct that there were contradictions in the captions and graphics. This was remiss of me, my apologies again. To correct this, I have reproduced figure 2 with the vector lengths explicitly shown and caption redone. This is shown overleaf in context.

The reviewer also correctly noted that I mention 9 subjects miscoded without specifying where this came from. It is indeed buried in the reference, but its introduction without explicitly saying this was jarring – I have now rephrased the section to read:

“This rendered the actual result highly fragile, given the demonstrable fact that inspection of the supplied tables in the paper in question demonstrated that at least 9 patients had been miscoded in the initial analysis.”

2. Thanks to the author for clarifying the definitions of Qe, Qc, and Qa (now defined as epsilon_E, etc). I especially appreciate the added sentence between equations (11) and (12) that interpret it for a non-statistical audience. However, I see that the Methods section has been moved after the Results and Discussion section. I am not strictly opposed to this decision; however, the result is that readers who read through the manuscript as it is written will see these novel technical terms used prior to their definitions. I suggest that the author either return to the ordering of sections that is more traditionally used for statistical articles and which was used in the original submission (Introduction, Methods, Results, Discussion) or add references to defining equations for any novel technical terms used at their first use and a simple, non-statistical interpretation.

I thank the reviewer for this observation, and agree it would be much easier to follow the flow if the typical ordering for such papers is allowed rather than putting the methodology at the end. I have redone the paper this way now. If *ELife* allow this, I agree it reads better – if not, I will tyr to re-write so that terms are defined in context. I am happy to chat with editorial team on this.

3. If the definitions /distinctions between x_i and x_m can be cleared up as per my first comment above, are the epsilons really necessary at all? Put differently, can the author please clarify why these statistics offer distinct information? Epsilon_A would seem to be just a rescaling of x_i by the sample size.

Hopefully my reply to point 1 and subsequent rewrite has clarified matters somewhat – my poor phrasing wrongly created a link between x_m_ and x_i_ which is hopefully clarified in this iteration.

4. Figure 4: I suggest to add a parenthetical that the FECKUP vector is in red. A legend such as what is used in Figure 2 would be helpful.

I whole-heartedly agree – figure has been re-rendered, and caption text modified as per points 5, 7 and 10.

5. Also, in the spirit of creating Figures that 'stand alone', can the caption of Figure 4 be modified to make clear which illustrative example it refers to? The only evidence that Figure 4 refers to the hypothetical liquid biopsy data is the use of the term 'city A' in the second sentence of the caption.

This is a very useful point, and I have modified all captions in light of this observation and the points raised in 7 and 10 after this. The new figure 4 caption text reads:

“Illustrative example 3 – An EOI analysis on the data supplied in the City A / City B screening comparison yields a FECKUP vector (in red) of 46.2 subjects, corresponding to a minimum tolerance of 66.5 total subjects after resolving the vector. As x_i_ = 73.7 (shown in green) with y_i_ = 62.7$ (shown in blue), but as the sensitivity and specificity of the tests used in city A are known, it can be shown that x_m_ ≈ 93, exceeding the limits of x_i_, placing the point within the ellipse and rendering any seeming significance void. Note that only a part of the ellipse of insignificance (denoted by the blue solid shape) is shown for clarity”

6. Figure 2 is not referenced anywhere in the text of the manuscript (as far as I can tell).

This is true, and is remedied now, thank you.

7. Can the author please add some sort of enumerative label or title to each of the illustrative examples and ensure that tables and figures make clear in their captions which illustrative example they refer to?

This has now been done for all figures and examples.

8. The third paragraph of the discussion says the method is 'only currently applicable to dichotomous outcome trials' but, assuming that the word 'trial' refers to an active intervention, I believe the author intends that this approach is more widely applicable to any study (that is, both interventional and observational) with a non-censored dichotomous endpoint.

This is absolutely true, and I have rewritten the discussion and introduction section to append the studies aspect, as I see this method having wider application in cohort studies and preclinical work.

9. I am still unclear on the need for the inequality in (13): does this inequality give a definition of epsilon_a, i.e. is epsilon_a really the entire set of numbers less than the RHS of this inequality? If so, why is epsilon_a only ever given as a single number and not an interval?

My apologies again, this was an artifact from a prior version – the equality is now there as it should have been previously.

10. Figure 4: presuming this Figure refers to the cancer screening example, the units of analysis here are more appropriately referred to as 'subjects' not 'patients'. This comment applies more broadly, i.e. the second paragraph of the discussion.

This is a very pertinent point – I have now corrected the manuscript to take this onboard in multiple areas.

11. The statement immediately following (15): "If xm >= xi or ym >= yi or both conditions are met…". Should there be absolute values around these expressions? For example, if xi < 0 and xm=0, then the sentence as currently written suggests that the results are not robust, which seems misleading. I believe the intended meaning here is "If |xm| >= |xi| or |ym| >= |yi| or both conditions are met…".

The reviewer is absolutely correct to note this – hopefully I have made it clearer that the only criteria that matter is as outlined in point 1 and 3, namely that the point (x_m_,y_m_) lies within the ellipse of insignificance.